# Chitosan Oligosaccharide Ameliorates Nonalcoholic Fatty Liver Disease (NAFLD) in Diet-Induced Obese Mice

**DOI:** 10.3390/md17070391

**Published:** 2019-07-02

**Authors:** Minyi Qian, Qianqian Lyu, Yujie Liu, Haiyang Hu, Shilei Wang, Chuyue Pan, Xubin Duan, Yingsheng Gao, Lian-wen Qi, Weizhi Liu, Lirui Wang

**Affiliations:** 1School of Basic Medicine and Clinical Pharmacy, State Key Laboratory of Natural Medicines, China Pharmaceutical University, Nanjing 211198, China; 2College of pharmacy and chemistry, Dali University, Dali 671003, China; 3MOE Key Laboratory of Marine Genetics and Breeding, College of Marine Life Sciences, Ocean University of China, Qingdao 266003, China; 4Laboratory for Marine Biology and Biotechnology, Qingdao National Laboratory for Marine Science and Technology, Qingdao 266235, China; 5School of Life Science and Technology, China Pharmaceutical University, Nanjing 211198, China; 6Clinical Metabolomics Center, China Pharmaceutical University, Nanjing 211198, China

**Keywords:** nonalcoholic fatty liver disease, chitosan oligosaccharide, lipidome, gut microbiome, tight junction

## Abstract

Nonalcoholic fatty liver disease (NAFLD) is a global epidemic, and there is no standard and efficient therapy for it. Chitosan oligosaccharide (COS) is widely known to have various biological effects, and in this study we aimed to evaluate the liver-protective effect in diet-induced obese mice for an enzymatically digested product of COS called COS23 which is mainly composed of dimers and trimers. An integrated analysis of the lipidome and gut microbiome were performed to assess the effects of COS23 on lipids in plasma and the liver as well as on intestinal microbiota. Our results revealed that COS23 obviously attenuated hepatic steatosis and ameliorated liver injury in diet-induced obese mice. The hepatic toxic lipids—especially triglycerides (TGs) and free fatty acids (FFAs)—were decreased dramatically after COS23 treatment. COS23 regulated lipid-related pathways, especially inhibiting the expressions of FFA-synthesis-related genes and inflammation-related genes. Furthermore, COS23 could alter lipid profiles in plasma. More importantly, COS23 also decreased the abundance of *Mucispirillum* and increased the abundance of *Coprococcus* in gut microbiota and protected the intestinal barrier by up-regulating the expression of tight-junction-related genes. In conclusion, COS23, an enzymatically digested product of COS, might serve as a promising candidate in the clinical treatment of NAFLD.

## 1. Introduction

Nonalcoholic fatty liver disease (NAFLD), a common liver disorder, is a global epidemic with an estimated prevalence of 25% worldwide [1,2,3]. NAFLD can progress from simple steatosis into nonalcoholic steatohepatitis (NASH), and in some cases to cirrhosis and even carcinoma [1,3,4,5]. NAFLD is characterized by extensive accumulation of hepatic triglycerides, which is closely associated with metabolic risk factors such as insulin resistance, type 2 diabetes mellitus, obesity, and abnormalities of lipid metabolism [5,6]. However, there is no current standard treatment for NAFLD [7,8]. Supplementation with antioxidants such as vitamin E and obeticholic acid has been reported to ameliorate NAFLD, but their long-term efficacy and safety is controversial [9,10]. With regard to pharmaceutical treatments, several drugs such as metformin, fibrates, statins, and pentoxifylline were reported to a show beneficial effect for NAFLD treatment, but there is limited clinical data supporting their efficacy, and safety concerns regarding these compounds have also been raised [7]. Therefore, there is a large unmet need to explore therapeutic approaches to ameliorate steatosis in livers and prevent NASH progression.

Recently, natural products have attracted extensive attention due to their safety and potentially beneficial effects [11]. Chitosan oligosaccharide (COS), the depolymerized product of chitosan, was reported to be non-toxic to humans and is widely used in the food and nutrition field [12,13]. COS’s small molecular size means that it can be optimally absorbed, and it contributes to systemic biological effects. COS has been reported to exhibit both preventive and curative effects against urinary bladder cancer progression in rodent models [14]. Also, previous study revealed the anti-diabetic effect of COS, and indicated that it could be used as pharmaceutical agent for the treatment of diabetes mellitus [15]. In addition, COS and its derivatives may offer therapeutic benefit in the treatment of Alzheimer’s disease by inhibiting β-secretase, acetylcholine esterase, neuronal apoptosis, and inflammatory responses [15]. More importantly, several studies have highlighted its role in anti-obesity in animal models [16,17,18], whereas the liver-protective effect of COS for NAFLD remains obscure.

Currently, there is an accumulation of evidence demonstrating the association between liver disease and intestinal microbiota [19,20]. Gut dysbiosis could result in the damage of intestinal mucosal barrier and the translocation of bacteria and microbial production, which will contribute to hepatic injury and inflammation [19,21]. COS and chitosan are reported to have potential applications as prebiotics for their beneficial influence on gut microbiota [22,23]. Here, we hypothesized that COS has a protective effect against liver disease via modulating gut microbiota. In this study, we obtained COS23, which was mainly composed of dimer and trimer, by enzymatically digesting chitosan, with the aim of evaluating its liver-protective effects in high-fat diet (HFD)-induced obese mice, and to explore the possible mechanism for this by detecting the influence of COS on gut microbiota.

## 2. Results

### 2.1. COS23 Alleviated Hepatic Steatosis in HFD-Induced Obese Mice

As shown in Figure 1A,B, the body weight and food intake of COS23-treated mice did not differ dramatically from those in the vehicle group (*p* > 0.05), suggesting that COS23 did not influence appetite. However, plasma alanine aminotransferase (ALT) was reduced significantly in COS23-treated mice compared with the vehicle group (Figure 1C), and plasma aspartate aminotransferase (AST) and alkaline phosphatase (ALP) were decreased in the COS23 group compared with the vehicle group (Figure 1D,E). Furthermore, the glucose tolerance test (GTT) results indicated that mice in the COS23-treated group were more sensitive to glucose stimulation, although no significant difference was observed (Figure 1F). Additionally, COS23-treated mice had decreased liver weight (*p* < 0.05), but the ratio of liver/body weight displayed no significant difference (Figure 1G). The two groups of mice had comparable adipose fat weight as well as a comparable ratio of brown adipose fat (BAT)/body weight and white adipose fat (WAT)/body weight (Figure 1G). The hematoxylin–eosin (H&E) and Oil Red O staining results revealed that lipid accumulation and hepatic steatosis were alleviated after COS23 treatment (Figure 1H,I). Altogether, COS23 effectively decreased liver weight, improved hepatic steatosis, and ameliorated plasma ALT, AST, and ALP levels in HFD-induced obese mice.

### 2.2. COS23 Modulated Hepatic Lipids

In order to investigate the influence of COS23 treatment on hepatic lipids, untargeted lipidomics were performed on the livers of mice from the vehicle and COS23-treated groups. We identified more than two hundred species of hepatic lipids (data not shown) belonging to 15 classes of lipids: carnitine (Car), ceramide (Cer), coenzyme Q9, diglyceride (DG), free fatty acid (FFA), lysophosphatidylcholine (LysoPC), lysophosphatidylethanolamine (LysoPE), monoglyceride (MG), phosphatidylcholine (PC), phosphatidylethanolamine (PE), phosphatidylglycerol (PG), phosphatidylinositol (PI), sphingomyelin (SM), sphingosine (Sph), and triglyceride (TG).

As presented in Figure 2A, a total of 40 species of lipids were observed to be significantly altered between the COS23-treated and vehicle groups. Among them, the alterations of seven species of TG (TG(49:4), TG(18:2/16:1/14:0), TG(18:2/16:0/14:0), TG(16:0/22:6/18:0), TG(16:0/20:5/22:1), TG(15:0/16:1/18:2), and TG(14:0/16:0/18:1)) were significantly different, and six species (except for TG(16:0/22:6/18:0)) were significantly decreased in the COS23-treated mice compared to the vehicle group. This was consistent with the results showing that total TG in livers were lower after COS23 treatment (Figure 2A,B), and also in line with H&E and Oil Red O staining. Moreover, six FFA species were significantly altered (*p* < 0.05), whose toxic effects on hepatocytes have been highlighted by previous studies [24]. Among these FFAs, all species were reduced in the livers of COS23-treated mice, especially FFA(16:0), FFA(16:1), and FFA(18:1) (Figure 2A), and the total hepatic FFAs in COS-treated mice were also decreased (Figure 2C). Meanwhile, other kinds of lipids belonging to groups SM, PC, PE, LysoPC, DG, coenzyme Q9, and Cer were also observed to be altered significantly after COS23 treatment. Overall, COS23 regulated hepatic lipids by altering the levels of several specific lipids.

### 2.3. COS23 Regulated Inflammation- and Lipid-Related mRNA Levels in HFD-Induced Obese Mice

To explore the possible mechanism of COS23’s effect, real-time PCR was adopted to evaluate the gene expression of FFA- and TG-related pathways. The expression of genes related to hepatic fatty acid synthesis such as ATP citrate lyase (*Acly*), fatty acid synthase (*Fasn*), and elongation of very long fatty acid (*Elovl5* and *6*) were significantly down-regulated in COS23-treated mice (Figure 3A). The levels of genes related to fatty acid β-oxidation such as carnitine palmitoyltransferase 1A (*Cpt1a*) and acyl-CoA oxidase 1 (*Acox1*) decreased without significant difference after COS23 treatment (Figure 3A). In terms of TG-related pathway genes, diacylglycerol O-acyltransferase 1, 2 (*Dgat1, 2*), patatin-like phospholipase domain containing 2 (*Pnpla2*), and lipase (*Lipe*) were also downregulated without significant difference following COS23 treatment (Figure 3B). We also found that inflammation-related genes such as C–C motif chemokine ligand2 (*Ccl2*) were inhibited in COS23-treated mice compared to the vehicle group, and the levels of other inflammation-related genes were also decreased after COS23 treatment, although no significant differences were observed (Figure 3C). Furthermore, the expression of collagen type I alpha 1 chain (*Col1a1*) was also lower in the COS23-treated group than in the vehicle group, confirmed by sirius red staining of liver sections, suggesting that hepatic collagen accumulation was improved by COS23. Overall, COS23 regulated lipid-related pathways; in particular, it inhibited the expression of FFA-synthesis-related and inflammation-related genes, and it also alleviated liver fibrosis in the HFD-induced obese mice.

### 2.4. COS23 Could Alter Lipid Profiles in Plasma

We next performed an un-targeted lipidomic analysis to assess the effect of COS23 on plasma lipids. A total of 143 lipid species were identified in plasma, and 23 of them showed significant differences between the COS23-treated and vehicle group mice (Figure 4A). Most lipids were decreased following COS23 treatment except PE(38:6), PE(36:4), PC(16:0/18:2), and cholesteryl ester (CE)(18:2). The total TG in plasma showed trends towards lower levels (*p* = 0.1097) in COS-treated mice compared to the vehicle mice (Figure 4B), as detected using a biochemical kit. In terms of alterations of levels of specific TGs, TG(54:7), TG(51:3), TG(18:2/16:0/16:0), and TG(14:0/18:1/18:2) were significantly decreased (*p* < 0.05) after COS23 treatment as revealed by the lipidome analysis (Figure 4A). Interestingly, 12 species of PCs in plasma were changed with a significant difference (*p* < 0.05) and most of them were reduced except for PC (16:0/18:2). Overall, our results showed that specific alterations in the plasma lipidome occurred following COS23 treatment, among which TG and PC levels changed the most.

### 2.5. COS23 Regulated Gut Microbiota Composition

16S rRNA gene sequencing analysis was performed to evaluate the influence of COS23 on gut microbiota. As revealed by α-diversity analysis, there was no significant difference in the Shannon and Simpson indexes between the vehicle and COS23-treated groups (Figure 5A,B). For the β-diversity, a principal component analysis (PCA) plot demonstrated a clearly distinct microbial landscape between vehicle and COS-treated mice (Figure 5C). At the phylum level, we found that the relative abundance of Deferribacteres dramatically reduced after COS23 treatment (Figure 5D). At the genus level, COS23 treatment induced a significant decrease of *Mucispirillum* and a marked increase of *Coprobacillus* (Figure 5E). Accordingly, we also found that the abundance of *Mucispirillum schaedleri* (*M. schaedleri*) was reduced after COS23 treatment (data not shown). According to the predictive functional clusters analyzed by PICRUSt, the most abundant functions were membrane transport and carbohydrate metabolism in the KEGG pathways (level 3), and 65 pathways were altered with significant differences after COS23 treatment (Appendix A). At the same time, we observed that the pathway of fatty acid biosynthesis was decreased significantly (*p* < 0.05) in COS23-treated mice. In contrast, fatty acid metabolism-related pathway was increased dramatically following COS23 treatment (*p* < 0.05). We further assessed the impact of COS23 on intestinal barrier function. As expected, the levels of tight-junction-related genes such as tight junction protein 1, 2 (*Tjp1*, *2*) in the distal small intestine (DSI) of COS23-treated mice were dramatically increased in comparison to the vehicle group, and the expression of Mucin 2 (*muc2*) and Occludin (*Ocln*) were up-regulated without significant difference. However, intestinal mRNA levels of antibacterial peptides, including regenerating family member 3 beta and gamma (*Reg3b* and *Reg3g*), exhibited no significant differences after COS23 treatment. Taken together, our results indicate that COS23 treatment could modulate gut microbiota and protect the intestinal barrier by up-regulating the expression of tight junction genes.

## 3. Discussion

NAFLD is a disease with global health implications with constantly increasing prevalence (15% in 2005 to 25% in 2010), and is strongly associated with the features of metabolic syndrome [25,26]. NAFLD covers a broad spectrum of liver diseases, and hepatic steatosis is considered to be the characteristic feature of NAFLD [27]. However, there are no standard and effective therapeutic strategies for NAFLD [7]. Natural products are promising sources to help develop new types of NAFLD drugs, and several natural products such as phenolics, flavonoids, alkaloids, and terpenoids have been reported to show beneficial effects on NAFLD [28]. COS, the natural polysaccharide hydrolyzed from shrimp shell chitosan, has attracted extensive attention because of its potential for use in various promising biomedical applications including anti-oxidation, anti-inflammation, immune-stimulation, and anti-hypertension [15,17,29]—all of which may contribute to a favorable effect on NAFLD. COS has also been widely studied as a potential candidate for the treatment of obesity [17,30,31], which is closely associated with liver disease; however, there are limited studies evaluating the effect of COS on NAFLD. Our results demonstrated that COS23 could significantly improve NAFLD in mice by decreasing hepatic lipids and inflammation with only a slight influence on body weight, which is consistent with the results of a previous study that showed that COS ameliorated lipid accumulation in palmitic-acid-induced HepG2 cells [32].

NAFLD is characterized by the accumulation of different lipid species in hepatocytes. The lipotoxicity of TG and FFA have been highlighted by several studies—particularly saturated fatty acids [24]—and as expected, we found that COS23 altered the hepatic lipid profile and significantly reduced FFA and TG in livers. Moreover, seven TG species and six FFA species were identified as having been dramatically altered after COS23 treatment. We also found that COS23 modulated lipid-related pathways; it especially inhibited the expression of FFA-synthesis-related genes such as *Fasn* and *Acly* as well as elongation of very long fatty acid (*Elovl5* and *6*). These results revealed that COS23 treatment decreased the accumulation of specific lipids in the classes of FFA and TG by regulating FFA- and TG-related pathways, thus contributing to the protection of hepatocytes from lipotoxic injury.

In addition, our results indicated that COS could lower plasma TG, which agrees with previous studies [30,33] and demonstrates that COS could play a favorable role in plasma lipid regulation by decreasing TG. However, no study has yet focused on the effect of COS on specific members of the TG family. The untargeted lipidome analysis in this study revealed that four species of lipids, including TG(54:7), TG(51:3), TG(18:2/16:0/16:0), and TG(14:0/18:1/18:2), were significantly reduced following COS23 treatment, indicating that these four species of TG may be responsible for the decreased total TG seen in plasma.

Currently, there is increasing evidence suggesting a link between gut microbiota and liver diseases [19,20], and the main mechanism for this is the translocation of intestinal microbiota, or their products, into portal circulation [21]. The intestinal barrier damage causing increasing intestinal permeability should be responsible for this translocation [21]; therefore, we performed 16S rRNA sequencing of the cecum content to investigate the influence of COS23 on gut microbiota as well as the intestinal barrier. As expected, the gut microbiota composition was greatly altered after COS23 treatment. The relative abundance of *Mucispirillum* genus was significantly reduced in COS-treated mice compared with the vehicle group mice. As reported, *Mucispirillum* is seen as a mucus inhabitant, and is widely associated with intestinal inflammation [34]. Furthermore, we observed that the relative abundance of *Coprococcus* was significantly increased after COS23 treatment. A recent human study reported that obese adults (BMI > 30 kg/m^2^) had a lower abundance of *Coprococcus* genus relative to the overweight group (25–29.9 kg/m^2^), suggesting that intestinal *Coprococcus* was negatively correlated with BMI and body fat percentage. Additionally, the expression of tight-junction-related genes increased after COS23 treatment. Thus, we speculated that COS23 modulated gut microbiota composition, particularly increasing *Coprococcus* and decreasing *Mucispirillum* abundance, which might prove beneficial to the intestine by improving mucosal barrier function.

Compared to chitosan or other polysaccharides, COS has a lower molecular weight and higher degrees of deacetylation and polymerization, which endow it with higher water solubility, lower viscosity, and enable it to be absorbed readily through the intestine [35]. Safety evaluations in mice revealed that its LD_50_ was greater than 10 g/kg in mice, suggesting a low toxicity of COS [36]. Although safety data in humans are still lacking, COS is widely used in the food and nutrition field and is widely known as non-toxic to the human body, making it a favorable candidate for clinical applications.

In conclusion, COS23—the enzymatically digested product of COS prepared in our study—efficiently attenuated hepatic steatosis by inhibiting FFA-synthesis-related genes, and improved hepatic inflammation as well as fibrosis. Our results showed that the mechanism for this may involve the regulation of gut microbiota and the improvement of intestinal barrier dysfunction by COS23. Although further clinical trials are needed, COS23 is a promising candidate in the clinical treatment of NAFLD.

## 4. Materials and Methods

### 4.1. Preparation of COS23

COS23 was prepared by the enzymatic degradation of chitosan according to our previous method [37,38]. Briefly, the solution including chitosan (8 mg/mL, 2.25 mL), chitosanase (0.25 mL), and sodium acetate buffer (0.2 M, pH 5.6, 2.5 mL) were incubated at 37 °C for 1800 min, shaking with a speed of 150 rpm. After hydrolysis, the resulting reaction was terminated by heating at 90 °C for 10 min, and the sample was lyophilized after centrifugation at 12,000× *g* for 10 min. Finally, the sample was examined by thin-layer chromatography.

### 4.2. Animal Experiments

Twenty male 5-week-old C57BL/6J mice were purchased from Beijing Vital River Laboratory Animal Technology Co., Ltd. (Beijing, China) and housed in the specific pathogen free (SPF) environment. After 11 weeks of high-fat diet (12492, 60% kcal from fat; Research Diets, New Brunswick, NJ, USA, the constitutions are listed in Appendix A), mice were randomly separated into vehicle and treatment groups (n = 10 each group). Mice in the control group (Vehicle) and COS-treated group (COS) drank water or COS (4% in drinking water) ad libitum during the experimental period. After another 10 weeks of HFD feeding, mice were sacrificed and samples including blood, liver and cecum were harvested. These samples were stored in −80°C after freezing in liquid nitrogen. All animal experiments were performed according to the guidelines of the China Pharmaceutical University Institutional Animal Care and Use Committee.

### 4.3. Determination of ALT, ALP, AST, and TG

The AST, ALT, and ALP levels in plasma, and the TG levels in plasma and liver were measured using commercial kits from Nanjing Jiancheng Bioengineering Institute (Nanjing, Jiangsu, China), performed as per manufactures’ instructions.

### 4.4. Glucose Tolerance Test (GTT)

After 13 weeks of HFD feeding, the GTT was performed according to the method described previously [39]. After 16 h of fasting, mice were injected intraperitoneally with glucose (1 g/kg) and glucose concentrations in blood were determined at the time points: 0, 15, 30, 60, 90 and 120 min.

### 4.5. Staining Procedures

Liver biopsies were fixed in formalin and embedded in paraffin, sectioned, stained with H&E [40], and scored according to NAFLD activity score (NAS). Frozen sections were stained with Oil Red O and analyzed by Image J to show hepatic lipid accumulation [40]. Liver sections for sirius red staining were performed as described previously [41].

### 4.6. Lipid Profiling

Untargeted lipidomes of the liver and plasma of mice from the vehicle and COS-treated groups were determined using liquid chromatography–mass spectrometry (LC–MS) using established methods [42,43,44]. Approximately 20 mg of liver tissue was transferred into homogenization tubes, followed by ice-cold dichloromethane/methyl alcohol (DCM/MeOH, 3:1, *v*/*v*) solution (50 μL of solvent/mg tissue). After the samples were homogenized with a beads-beater (Biospec Products, Inc., Bartlesville, OK, USA), 300-μL aliquots of distilled water were added. The solution was vortexed (10 min) and centrifuged (13,000 rpm for 10 min at 4 °C), then the DCM layer was transferred, evaporated, and finally reconstituted for LC–MS analysis (positive and negative modes) [42,43]. For plasma, 300 μL of ice-cold MeOH was mixed with plasma (50 μL) and vortexed thoroughly. Afterwards, methyl tert-butyl ether was added, followed by the addition of distilled water (300 μL). The supernatant was finally evaporated and reconstituted for LC–MS analysis [43,44].

Subsequently, lipids were separated using a reverse-phase UPLC ACQUITY C8 BEH column (2.1 mm × 100 mm × 1.7 μm, Waters, Milford, MA, USA). A 1290 Infinity Ultra-High-Performance Liquid Chromatography (UHPLC) System coupled to a 6545 quadrupole time-of-flight mass spectrometer (LC-QTOF-MS) (Agilent Technologies, Chandler, AZ, USA) equipped with a Dual Agilent Jet Stream electrospray source was used for analysis. The system was controlled by the MassHunter Data Acquisition Software (Version B.06.01, Agilent Technologies, Chandler, AZ, USA).

The raw data files generated were converted to mzData format through a DA reprocessor, and 2000 counts were used as threshold. Then, XCMS software (R-3.3.3, http://metlin.scripps.edu/download/) was used to accomplish peak finding, filtering, and alignment. Lipid identification was performed based on MS/MS fragments, and the accurate mass was obtained using the online lipid database of Lipid Maps (www.lipidmaps.org) [45].

### 4.7. Real-Time PCR

RNAiso Plus (TaKaRa, Dalian, Liaoning, China) was used to extract total RNA from livers and intestinal tissues. cDNA was synthesized from 1 μg of RNA using a High-Capacity cDNA reverse transcription kit (Applied Biosystems, Foster City, CA, USA). Real-time PCR was performed to amplify the cDNA products using the SYBR Premix (Bio-Rad, Hercules, CA, USA). All real-time PCR primer sequences were designed using Primer3Plus (http://www.primer3plus.com/). The relative expression levels of targeted genes were normalized to 18S.

### 4.8. 16S rRNA Gene Sequencing and Analysis

Cecum contents were collected and the bacterial genomic DNA extracted using established methods described before [40]. The V1–V3 region of 16S rRNA gene was amplified and sequenced on the Illumina MiSeq platform (Illumina, San Diego, CA, USA) in Annoroad Gene Technology Co., Ltd (Beijing, China). The resulting sequences were analyzed by QIIME2 workflows [46]. Then, PICRUSt was adopted to predict the functional composition profiles through the KEGG (Kyoto Encyclopedia of Genes and Genomes) pathway database [47], which was eventually analyzed by the Statistical Analysis of Metagenomic Profiles (STAMP) software (v2.1.3, http://kiwi.cs.dal.ca/Software/STAMP).

### 4.9. Statistical Analysis

For the 16S rRNA results, the significance of the differential abundance at the genus level between the vehicle and COS-treated groups were estimated using a one-way ANOVA with post hoc Tukey’s test. The significance of the differences in the lipidome results were analyzed using a Student’s *t*-test, and for other results, the differences between the two groups were analyzed using a Mann–Whitney test (GraphPad PRISM 5). All data were represented as mean ± SEM and a *p*-value < 0.05 was considered statistically significant.

## Figures and Tables

**Figure 1 marinedrugs-17-00391-f001:**
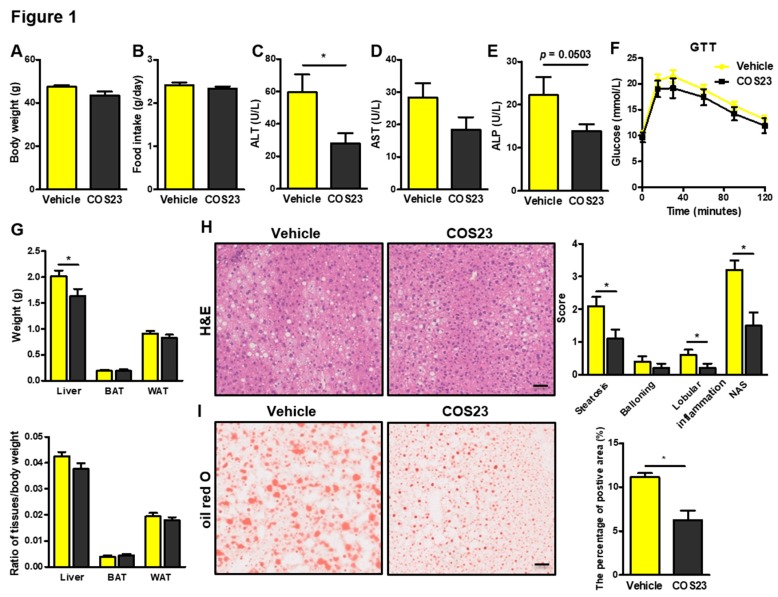
COS23 (an enzymatically digested product of chitosan oligosaccharide (COS)) improved liver steatosis in diet-induced obese mice. (**A**) Body weight and (**B**) food intake. Plasma (**C**) alanine aminotransferase (ALT), (**D**) aspartate aminotransferase (AST), and (**E**) alkaline phosphatase (ALP). (**F**) Blood glucose levels in the glucose tolerance test (GTT). (**G**) Liver and adipose weight (upper); ratio of tissue/body weight (lower). (**H**) Representative images of hematoxylin–eosin (H&E) staining and the histological score of liver sections according to nonalcoholic fatty liver disease (NAFLD) activity score (NAS—the sum of the scores for steatosis (0–3), lobular inflammation (0–3), and ballooning (0–2), ranging from 0 to 8). (**I**) Oil Red O staining of liver sections of obese mice after vehicle and COS23 (4%) treatment, quantitated by image analysis (Scale bar: 50 μm). Data are represented as mean ± SEM (n = 10). * *p* < 0.05.

**Figure 2 marinedrugs-17-00391-f002:**
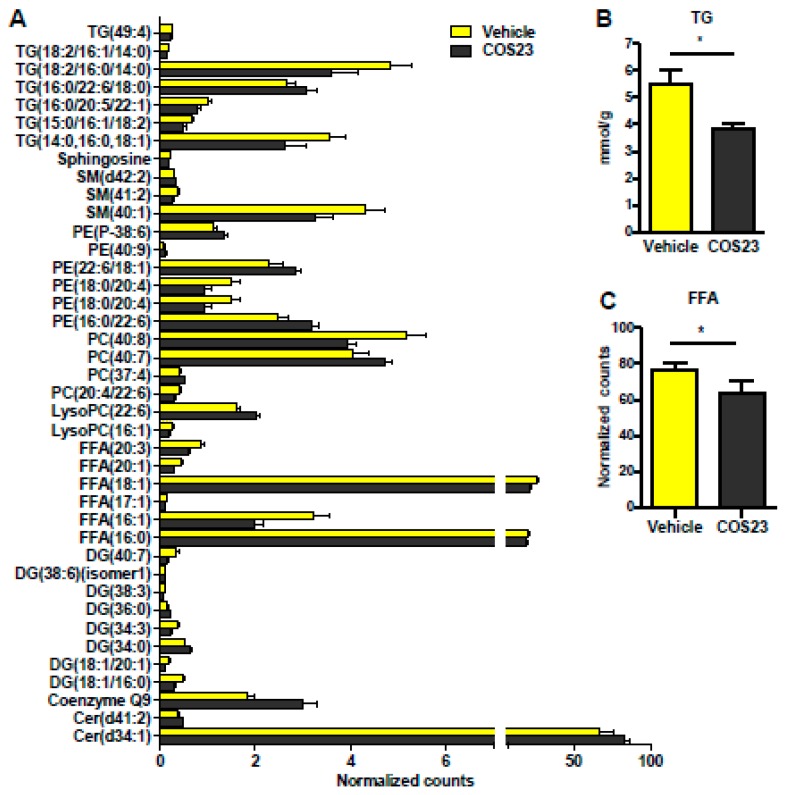
COS23 modulated hepatic lipids. (**A**) Identification of 40 species of lipids in the livers with significant difference between mice in COS23-treated and vehicle groups. (**B**) Total hepatic triglycerides (TGs) detected by a biochemical kit. (**C**) Total hepatic free fatty acids (FFAs). Data are represented as mean ± SEM (n = 10). * *p* < 0.05. Cer: ceramide; DG: diglyceride; LysoPC: lysophosphatidylcholine; PC: phosphatidylcholine; PE: phosphatidylethanolamine; SM: sphingomyelin.

**Figure 3 marinedrugs-17-00391-f003:**
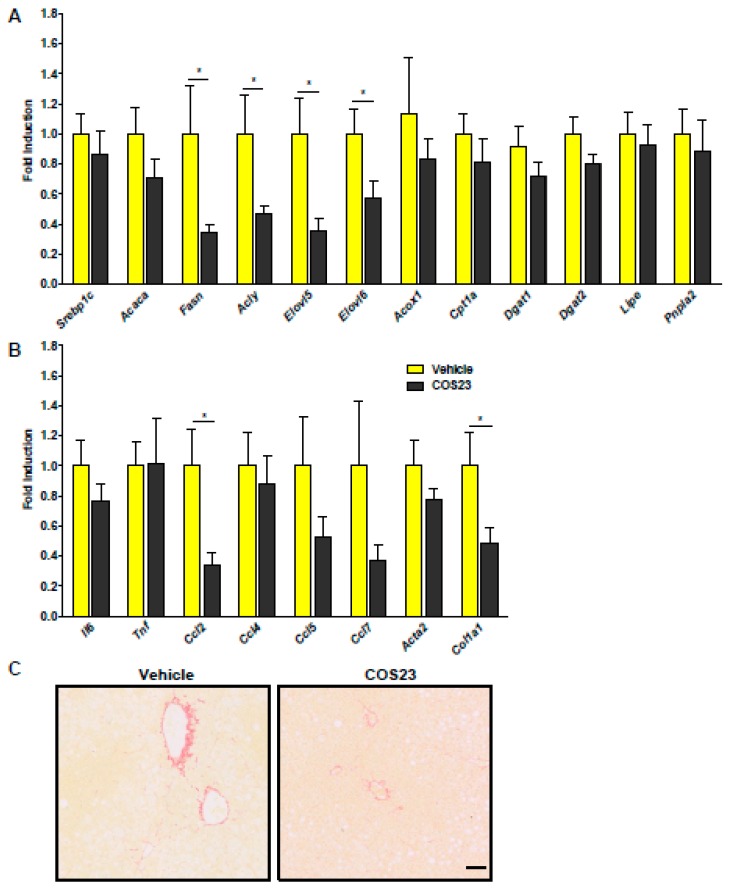
COS23 regulated the expression of hepatic lipid and inflammation-related genes. (**A**) mRNA levels of FFA and TG biosynthesis- and hydrolysis-related genes. (**B**) The expression levels of inflammation-related genes. Data are represented as mean ± SEM (n = 10). * *p* < 0.05. (**C**) Representative images of liver sections stained by sirius red. Scale bar: 50 μm.

**Figure 4 marinedrugs-17-00391-f004:**
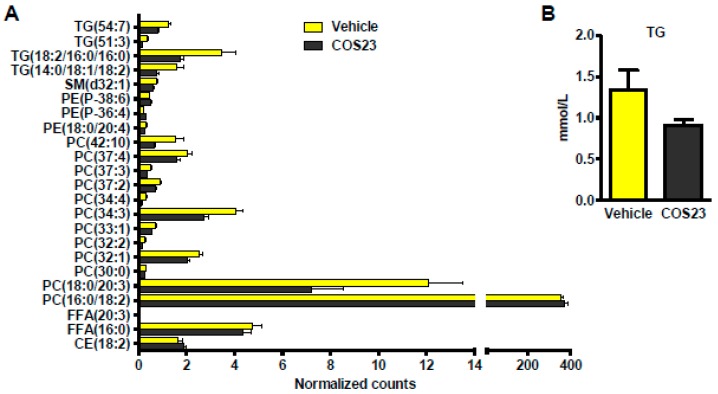
COS23 modulated lipids in plasma. (**A**) Twenty-three species of lipids in plasma were altered significantly following COS23 treatment. CE, cholesteryl ester. (**B**) TG in plasma determined by a biochemical kit. Data are represented as mean ± SEM (n = 10). * *p* < 0.05.

**Figure 5 marinedrugs-17-00391-f005:**
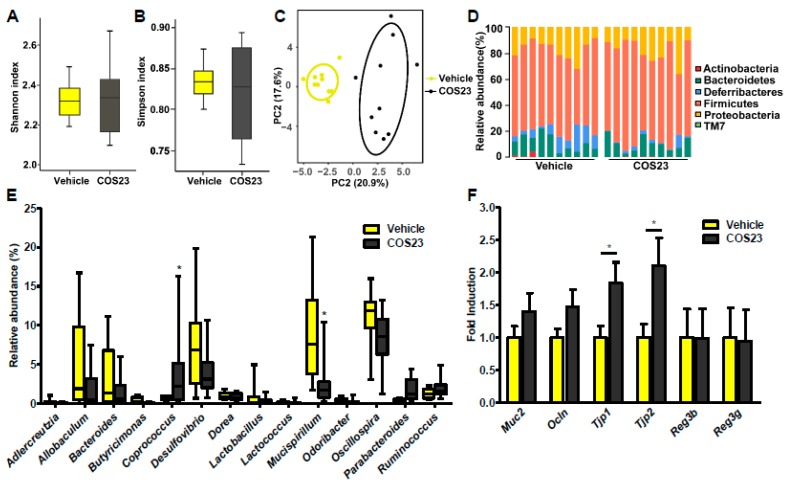
COS23 altered gut microbiota composition and improved intestinal barrier. (**A**,**B**) α-diversity in diet-induced obese mice after vehicle and COS23 treatment. (**C**) Principal component analysis (PCA) of gut microbiota between mice in vehicle and COS23 treatment groups. (**D**) The relative abundance of gut microbiota at the phylum level. (**E**) The relative abundance of gut microbiota at the genus level (top 14 genera). (**F**) Intestinal mRNA levels of tight-junction- and antibacterial-peptide-related genes. Data are represented as mean ± SEM (n = 10). * *p* < 0.05.

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
