# Peer review of "Chitosan Oligosaccharide Ameliorates Nonalcoholic Fatty Liver Disease (NAFLD) in Diet-Induced Obese Mice"

_marinedrugs, 2019, doi:10.3390/md17070391_

Round 1
Reviewer 1 Report
In this study, the authors describe the function of chitosan oligosaccharide (COS - named COS23) in the clinical treatment of Nonalcoholic fatty liver disease (NAFLD). The authors demonstrated that COS23 reduced hepatic steatosis and improved liver injury in diet-induced obese mice.
Overall, the manuscript is written well. However, some improvements are necessary, especially in the introduction and discussion sections to obtain a broad readership. The introduction is not informative; the authors need to be included the background on the aspect of COS in relation to other drug discovery reports. Also, the discussion section is too long but without including key points of COS and Nonalcoholic fatty liver disease from the literature.
Is chitosan oligosaccharide extracted from any marine organisms? If so, the authors should include the info about that marine source in the manuscript and should elaborate on the procedure of extractions.
Author Response
Comments and Suggestions for Authors
Point 1: In this study, the authors describe the function of chitosan oligosaccharide (COS-named COS23) in the clinical treatment of Nonalcoholic fatty liver disease (NAFLD). The authors demonstrated that COS23 reduced hepatic steatosis and improved liver injury in diet-induced obese mice. Overall, the manuscript is written well. However, some improvements are necessary, especially in the introduction and discussion sections to obtain a broad readership. The introduction is not informative; the authors need to be included the background on the aspect of COS in relation to other drug discovery reports.
Response 1: Thank you so much for your careful review, and we feel so grateful for your suggestions and comments, which help us a lot to improve this manuscript. We have revised each specific comment and all the changes have been highlighted in the modified version of the manuscript. We have added more background of COS in the introduction, please see lines 59-65 in the revised manuscript.
Point 2: Also, the discussion section is too long but without including key points of COS and Nonalcoholic fatty liver disease from the literature.
Response 2: Thank you so much for your comment and suggestion. We have added more information about COS and nonalcoholic fatty liver disease in the discussion (lines 200-204; lines 206-207).
Point 3: Is chitosan oligosaccharide extracted from any marine organisms? If so, the authors should include the info about that marine source in the manuscript and should elaborate on the procedure of extractions.
Response 3: Many thanks for your valuable advice. COS23 was prepared by enzymatic degrading of chitosan, and we have added more details of the procedures of COS23 extractions (lines 263-266).
Reviewer 2 Report
Comments to the Editor
Minyi Qian investigated the effect of Chitosan oligosaccharide on nonalcoholic fatty liver disease in HFD-induced obese mice. This original paper encompasses an interesting topic and may be worthwhile publishing, however it needs some revisions.
Comments:
1. Authors didn’t assess the volume of oral intakes. Chitosan oligosaccharide might repress oral intakes. Pair feeding is best way to assess the direct effect of Chitosan oligosaccharide except for appetite loss.
2. Authors could estimate the function of gut microbiota using with KEGG. Authors should examine whether lipid metabolism in gut microbiota was altered by Chitosan oligosaccharide.
3. Authors should assesse the historical grade of NAFLD using NAFLD activity score.
4. Authors should evaluate oil-red-O stained area.
5. Please provide the information of constitutions of HFD.
6. Line 266, 4.5. Staining procedures: I guess authors didn’t perform liver biopsy in this study.
Author Response
Dear reviewer,
Many thanks for your positive comments and helpful suggestions to this manuscript, which help us a lot to improve this manuscript. We have revised each specific comment and all the changes have been highlighted in the modified version of the manuscript.
Comments to the Editor
Minyi Qian investigated the effect of Chitosan oligosaccharide on nonalcoholic fatty liver disease in HFD-induced obese mice. This original paper encompasses an interesting topic and may be worthwhile publishing, however it needs some revisions.
Comments:
Point 1: Authors didn’t assess the volume of oral intakes. Chitosan oligosaccharide might repress oral intakes. Pair feeding is best way to assess the direct effect of Chitosan oligosaccharide except for appetite loss.
Response 1: Thank you so much for your careful review. Unfortunately, we did not record the volume of oral intakes of COS23 every weeks in view of the fact that we realized that the mice in COS23-treated group drink the similar volume of water (4% of COS23) to the vehicle group. So we could not list the precise oral intake of COS23. During the experiment, we recorded the food intake every week, and the mice in vehicle COS23-treated group consumed similar food, suggesting COS23 did not influence appetite. We have added the information in the revised manuscript. (Lines 79-80)
Point 2: Authors could estimate the function of gut microbiota using with KEGG. Authors should examine whether lipid metabolism in gut microbiota was altered by Chitosan oligosaccharide.
Response 2: Thank you so much for your valuable advice. We adopted PICRUSt to predict functional composition profiles through KEGG pathway database, the related information has been added in the new manuscript. (Line177-183; Supplementary Materials Figure S1)
Point 3: Authors should assesse the historical grade of NAFLD using NAFLD activity score.
Response 3: Many thanks for your valuable advice. We have added the historical score in the revised manuscript (Figure 1).
Point 4: Authors should evaluate oil-red-O stained area.
Response 4: Thanks a lot for your kind suggestion. We have added the figure of oil-red-O stained area (Figure 1).
Point 5: Please provide the information of constitutions of HFD.
Response 5: Thanks very much for your advice. The constitutions of HFD has been added in the Supplementary Materials Table S1.
Point 6: Line 266, 4.5. Staining procedures: I guess authors didn’t perform liver biopsy in this study.
Response 6: Thank you very much for your careful review. The liver biopsies were stained with H&E, oil red O and sirius red, and the related methods were described in the manuscript. (Line 290-293)
Round 2
Reviewer 2 Report
Authors answered all comments, which I provided, and revised the manuscript well.